# Effectiveness of Cell-Free and Concentrated Ascites Reinfusion Therapy in the Treatment of Malignancy-Related Ascites: A Systematic Review and Meta-Analysis

**DOI:** 10.3390/cancers13194873

**Published:** 2021-09-29

**Authors:** Hao Chen, Masashi Ishihara, Nobuyuki Horita, Shigeru Tanzawa, Hiroki Kazahari, Ryusuke Ochiai, Takahiko Sakamoto, Takeshi Honda, Yasuko Ichikawa, Kiyotaka Watanabe, Nobuhiko Seki

**Affiliations:** 1Division of Oncology, Department of Internal Medicine, Teikyo University School of Medicine, Tokyo 173-8606, Japan; chinsmd@gmail.com (H.C.); m.ishihara@med.teikyo-u.ac.jp (M.I.); s.tanzawa@med.teikyo-u.ac.jp (S.T.); kazahari0309@med.teikyo-u.ac.jp (H.K.); ryo7132003@med.teikyo-u.ac.jp (R.O.); tsakamoto@med.teikyo-u.ac.jp (T.S.); thonda@med.teikyo-u.ac.jp (T.H.); icchi@med.teikyo-u.ac.jp (Y.I.); kiyowata@med.teikyo-u.ac.jp (K.W.); 2Department of Pulmonology, Yokohama City University Graduate School of Medicine, Yokohama 236-0004, Japan; horitano@yokohama-cu.ac.jp

**Keywords:** cell-free and concentrated ascites reinfusion therapy, malignant-related ascites, chemotherapy

## Abstract

**Simple Summary:**

Cell-free and concentrated ascites reinfusion therapy (CART) was a safe and effective palliative therapy in malignancy-related ascites. Abdominal distension, dyspnea, and fatigue were alleviated significantly after CART. The mean time to the next paracentesis was 20.7 days. In total, 17% of patients had improved performance status after CART.

**Abstract:**

Background: Malignancy-related ascites (MRA) is one of the symptoms causing discomfort in advanced cancer patients. Cell-free and concentrated ascites reinfusion therapy (CART) is one of the palliative treatments widely conducted in Japan only. Methods: A systematic review following a meta-analysis of CART was performed. The efficiency and adverse events were evaluated. Results: A total of 2567 patients and 6013 procedures of CART were identified in this study. The mean volume of MRA collected was 4.29 (95% confidence interval (CI) 3.47–5.11) L, and the volume reinfused after concentrating was 0.49 (95% CI 0.39–0.60) L. A total of 86.1 (95% CI 77.1–95.2) g protein and 42.9 (95% CI 36.0–50.0) g albumin was reinfused. The mean time to the next paracentesis was 20.7 (95% CI 15.6–25.8) days. The body weight was reduced by 3.38 (95% CI 1.90–4.86; *p* < 0.01) kg, and abdominal circumference was reduced by 7.86 (95% CI 6.58–9.14; *p* < 0.001) cm. Serum albumin increased an average of 0.14 (95% CI −0.01–0.28; *p* = 0.07) mg/dL the day after CART. Abdominal distension, dyspnea, and fatigue were alleviated by 6.0 (95% CI 5.59–6.51), 2.66 (95% CI 2.05–3.28), and 2.64 (95% CI 1.86–3.42) points using a numerical rating scale system ranging from 0 to 10. Overall, 17% (95% CI 0.03–0.31%) of patients had improved performance status after CART. Significant body temperature elevation was observed, at an average of 0.4 °C (95% CI 0.18–0.62 °C). Conclusions: CART might be a safe and effective palliative therapy in MRA and further clinical trials are necessary.

## 1. Introduction

Ascites is the pathological accumulation of fluid within the abdominal cavity. The most common causes of malignancy-related ascites (MRA) are adenocarcinomas of the ovary, which account for approximately 10% of all cases of ascites, followed by carcinomas of the breast, colon, stomach, and pancreas [1]. MRA results in impairment in quality of life (QOL) and significant symptoms, mainly due to increased intraabdominal pressure and pain, nausea, anorexia, vomiting, fatigue, and dyspnea [2]. It is also a sign of advanced cancer and a poor prognosis, averaging about 20 weeks from time of diagnosis [3].

A variety of medical, interventional, and surgical therapies are now available for the management of both complications and symptoms, but there are limited guidelines for the treatment of MRA [4,5,6]. The most common first-line options are diuretics and intermittent paracentesis. Even small-volume paracentesis can alleviate the abdominal distension of terminally ill cancer patients with malignant ascites [7]. Indwelling catheters, ports, and shunts have all been proposed to reduce morbidity and improve QOL [8]. A newer modality, hyperthermic intraperitoneal chemotherapy (HIPEC), has demonstrated a survival advantage as a prophylactic strategy in gastric and ovarian cancers [9,10].

In palliative therapies, cell-free and concentrated ascites reinfusion therapy (CART) is also used for treating refractory ascites. CART comprises three processes. After the ascites is first filtered to remove cell components, it is concentrated to reduce its volume. The fluid obtained through these processes, including useful proteins such as albumin and globulin, is finally reinfused intravenously [11]. This therapy has been widely used in Japan to reduce symptoms in patients with MRA. Hypoalbuminemia, which is prone to malignant ascites, has been reported as a risk factor for febrile neutropenia, a side effect of cancer chemotherapy [12]. With CART, the proteins included in ascites are collected and intravenously reinfused, avoiding the loss of beneficial proteins through paracentesis.

There was no controlled trial conducted between CART and puncture groups to determine the efficacy and safety of CART. To explore the efficacy and safety of CART for malignant ascites, we conducted a systematic review and meta-analysis of observational studies.

## 2. Materials and Methods

### 2.1. Study Overview

The protocol of this systematic review and meta-analysis was composed following the standard guidelines for a systematic review of the Preferred Reporting Items for Systematic Reviews and Meta-Analyses (PRISMA) statement and registered on the website of the University Hospital Medical Information Network Clinical Trials Registration (UMIN000044541) [13,14]. Institutional Review Board approval was not required because of the nature of this study.

### 2.2. Study Search

Four major online databases, namely PubMed, Web of Science, Cochrane, and Embase, were searched. The following formula was applied for PubMed: (ascites reinfusion therapy) OR (cell-free and concentrated ascites reinfusion therapy) OR (CART) AND (malignant ascites). Two review authors (MI and HC) independently screened the titles and abstracts and carefully evaluated the full text to select eligible articles. In cases of discrepancy, they reached a consensus through discussion. Review articles and the included original articles were hand-searched (MI and HC) for additional research papers that met the inclusion criteria.

### 2.3. Inclusion and Exclusion Criteria

Full articles and brief reports published in any language that provided data for the effectiveness of CART for malignant ascites were examined. To be included, a study had to include (1) patients with malignant ascites, (2) data evaluating the effectiveness of CART, and (3) detailed information of the CART procedure. The exclusion criteria were as follows: (1) data of procedure only, (2) cirrhosis ascites only, and (3) no information identifying the efficacy of CART.

### 2.4. Risk of Bias

Two reviewers independently assessed the methodological quality of the selected studies using the Newcastle–Ottawa quality assessment, evaluating the quality of observational studies [15]. Due to the nature of single-arm studies, “selection of the non-exposed cohort” and “comparability” domains were not applicable in this study. The final score would then be 6 stars at most in the modified Newcastle–Ottawa quality assessment. Disagreement between reviewers was discussed, and agreement was reached by consensus.

### 2.5. Outcomes

The general characteristics of CART, such as the volume of ascites collected and reinfused, were analyzed. The efficacy of CART was identified by reduced body weight and abdominal circumference, increased serum albumin and total protein, and improved estimated glomerular filtration rate (eGFR) and creatinine. Eastern Cooperative Oncology Group performance status (ECOG PS) and alleviation of symptoms including abdominal distension, dyspnea, fatigue, lack of appetite, abdominal pain, and nausea and vomiting were evaluated [16]. The adverse events of CART were evaluated using the common terminology criteria for adverse events (CTCAT ver. 5.0).

Laboratory findings were checked the day after each CART procedure and adverse events were appraised during the procedure of CART. The ratio of improved PS was considered as the primary outcome of the long-term effect of CART, but only four studies discussed PS as a category variable. A short-term effect of CART was analyzed by the changes in body weight, abdominal girth, and serum albumin according to the paracentesis and reinfusion procedures in CART.

### 2.6. Data Extraction

Two review authors, MI and HC, independently extracted data, including the name of the first author, the publication year, the publication country, the types of immunohistochemical markers, the numbers of patients with positive results, the numbers of patients evaluated, and Newcastle–Ottawa quality assessment-related information.

### 2.7. Statistics

All analyses were performed in Review Manager ver. 5.3 (Cochrane Collaboration, Oxford, UK). Figures prepared using Review Manager were adjusted as necessary. Mean differences and 95% confidence intervals (95% CIs) were compared before and after CART. Heterogeneity evaluated with the I^2^ statistic was interpreted as follows: I^2^  =  0% indicates no heterogeneity, 0%  <  I^2^  <  25% indicates the least heterogeneity, 25%  ≤  I^2^  <  50% indicates mild heterogeneity, 50%  ≤  I^2^  <  75% indicates moderate heterogeneity, and 75%  ≤  I^2^ indicates strong heterogeneity [17]. A *p*-value of < 0.05 was considered significant.

## 3. Results

### 3.1. Study Search and Study Characteristics

A total of 95 articles, including 93 articles through database searching and 2 articles by hand-searching, were identified. There were 63, 45, and 15 articles left after removing duplication, screening, and full-article reading, respectively (Appendix A). There was no consensus on the direction of filtration, that is, inside–out or outside–in, and the processing conditions, such as filtration speed, concentration speed, and driving force to filter and concentrate ascites. Keisuke-modified cell-free and concentrated ascites reinfusion therapy (KM-CART) was used in 2128 patients, with an outside–in filtration direction. All studies used the same AHF-MO ascitic filtration filter and AHF-UP ascitic concentration filter (both from Asahi Kasei Medical Co., Ltd., Tokyo, Japan) for filtering and concentrating the ascites. The outcomes of the enrolled studies are shown in Appendix A. Due to the nature of the single-arm study, the modified Newcastle–Ottawa Scale scores varied from 4 to 6 stars (Appendix A).

A total of 2567 patients with MRA and 6013 procedures of CART were identified [18,19,20,21,22,23,24,25,26,27,28,29,30,31,32] (Table 1). All studies were conducted in Japan. All but two articles were reported in English [22,27]. There were four and five studies focused on MRA due to gastric and gynecological cancers, respectively, and the remaining six studies enrolled MRA patients with different kinds of malignancies. Chemotherapy with CART was reported by seven studies. The mean volume of MRA collected was 4.29 L (95% confidence interval (CI) 3.47–5.11 L), and the volume reinfused after concentrating was 0.49 L (95% CI 0.39–0.60). A total of 86.1 g (95% CI 77.1–95.2 g) protein and 42.9 g (95% CI 36.0–50.0 g) albumin was reinfused. The mean time to the next paracentesis was 20.7 days (95% CI 15.6–25.8 days) (Appendix A).

### 3.2. Efficiency of CART

After CART, mean body weight was reduced by 3.38 kg (95% CI 1.90–4.86 kg; *p* < 0.01; I^2^ = 0%, *p* for heterogeneity = 0.98) (Figure 1), and abdominal circumference was reduced 7.86 cm (95% CI 6.58–9.14 cm; *p* < 0.01; I^2^ = 12%, *p* for heterogeneity = 0.34) (Figure 2). The day after CART, serum albumin increased an average of 0.14 mg/dL (95% CI −0.01–0.28 mg/dL; *p* = 0.07; I^2^ = 91%, *p* for heterogeneity <0.01) (Figure 3), and serum total protein increased 0.18 mg/dL (95% CI −0.23–0.59 mg/dL; *p* = 0.39; I^2^ = 98%, *p* for heterogeneity <0.01) (Appendix A). Only one study reported decreased concentrations of protein and albumin due to the infusion of 500 mL to 1500 mL during the procedure of paracentesis to prevent hypotension [25]. Creatinine was decreased 0.1 g/dL (95% CI 0.07–0.13 g/dl; *p* < 0.01; I^2^ = 0%, *p* for heterogeneity = 0.94) (Appendix A), and eGFR improved 6.95 mL/min/1.73 m^2^ (95% CI 5.37–8.54 mL/min/1.73 m^2^; *p* < 0.01; I^2^ = 98%, *p* for heterogeneity= 0.68) (Appendix A).

An improvement in PS was reported by 10 studies. Due to different evaluation methods, only four studies were included in the meta-analysis, showing that 17% (95% CI 3–31%) of the patients had improved PS after CART [24,26,27,29] (Appendix A). Most studies also described the alleviation of symptoms after CART, with abdominal distension, dyspnea, and fatigue improved 6.0 (95% CI 5.59–6.51), 2.66 (95% CI 2.05–3.28), and 2.64 (95% CI 1.86–3.42) points using a numerical rating scale system ranging from 0 to 10 (Table 2) [18,25].

### 3.3. Adverse Events in CART

Only two studies reported hypotension during ascites drainage [19,25]. Nine studies used 100 or 200 mg of hydrocortisone before reinfusion to prevent fever during reinfusion, but an increase in body temperature was observed, at an average of 0.4 °C (95% CI 0.18–0.62 °C) (Appendix A). There were only four episodes of Grade 2 fever according to the common terminology criteria for adverse events (CTCAE). A decrease in platelets was also observed after CART 4.39 × 10^4^/μL (95% CI 2.22–6.57 × 10^4^/μL; *p* < 0.01; I^2^ = 60%, *p* for heterogeneity = 0.01) (Appendix A).

## 4. Discussion

This study evaluated the effects of CART in 2567 patients and 6013 procedures, with a mean time to next paracentesis of 20.7 days, showing that CART reduced body weight and abdominal circumference, increased serum albumin and total protein, improved PS, and alleviated symptoms. Significant body temperature elevations, by 0.4 °C on average, were observed among the patients, although these were not clinically important. This was reported by previous studies [19]. In addition, decreased platelets were observed after CART, but there was no report on the necessity of platelet transfusion. Only limited adverse events were observed during the procedures, except for four episodes of Grade 2 fever (CTCAE Ver5.0). CART was identified as a safe and effective palliative therapy in the treatment of MRA.

Fever was a notable adverse event upon the reinfusion of concentrated ascites. Concomitant steroid and/or non-steroidal anti-inflammatory drug (NSAID) use before reinfusion was significantly and negatively associated with increases in body temperature [19]. The concentration of inflammatory cytokines in ascites was not related to body temperature change, and the presence of IL-10 in ascites was related to longer survival after CART [21]. Of special note was that albumin, total protein, and eGFR were significantly increased after the reinfusion of concentrated ascitic fluid. It is considered that symptom relief itself can be achieved by paracentesis alone without the reinfusion of collected ascites. The effects of reinfusion of the concentrated ascitic fluid may be maintained for 20.7 days, this potentially being longer than the effects of total paracentesis alone (10 to 14 days) [33,34,35].

In total, 17% of patients showed increased ECOG PS after CART, and continued chemotherapies were frequently conducted in these patients. The combination of CART and antineoplastic agents was proven to be as safe as CART alone in cases of MRA [36]. CART may contribute to improved survival in patients with advanced gynecological and gastrointestinal cancers [26,29]. The combination of CART followed by chemotherapy could be a treatment option for cancer patients with MRA.

Several limitations to this study must be considered when interpreting the results. First, there were no controlled trials, and all studies included in this review were single-arm studies. Second, there was no consensus about puncture volume, concentrate ratio, etc., in the procedure of CART, resulting in high heterogeneity in most subgroup analyses. Third, the background characteristics of patients enrolled in the study might be different due to the nature of an observational study, which posed a substantial risk for selection bias. Fourth, the limited study investigated the long-term effect of CART. Although various outcomes were compared before and just after the procedure, they seemed to lack reliability for the long-term efficacy of CART. Fifth, the effect of CART was analyzing per procedure, which posed a substantial risk of selection bias. Sixth, CART was used in Japan only, and whether the conclusion can be generalized to different countries needs to be verified.

## 5. Conclusions

CART might be a safe and effective palliative therapy for MRA. There were no serious adverse events during the procedures. Further clinical trials are needed to confirm the efficacy and safety of CART for malignant ascites.

## Figures and Tables

**Figure 1 cancers-13-04873-f001:**
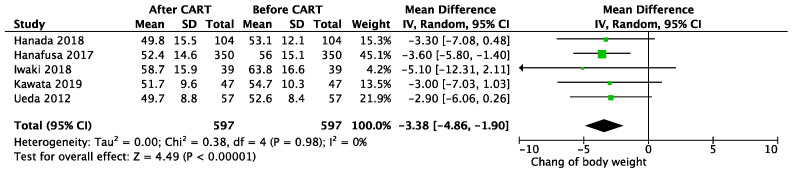
Change in body weight before and after CART.

**Figure 2 cancers-13-04873-f002:**
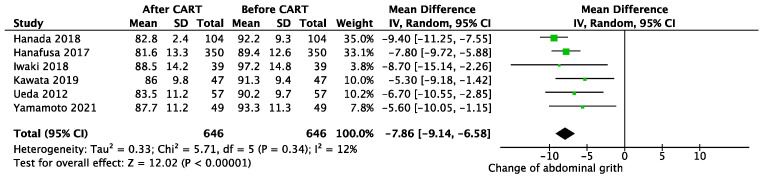
Change in abdominal circumference before and after CART.

**Figure 3 cancers-13-04873-f003:**
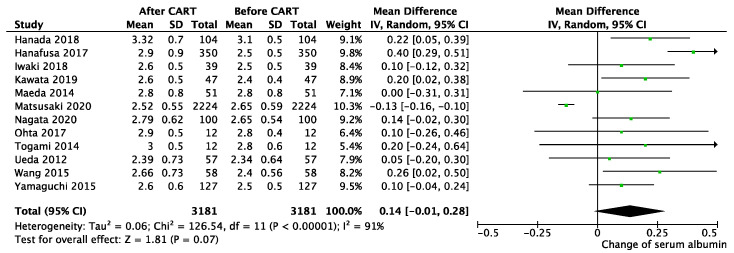
Change in serum albumin before and after CART.

**Table 1 cancers-13-04873-t001:** Background and characteristics of studies included.

First Author	Year	Tumors	Types	Patients	Chemotherapy	Age (y)	Procedures	Collection (mL) (SD)	Reinfusion (mL) (SD)	Protein (g) (SD)
Hanada	2018	mixed	CART	51	14	64	104	5855 (1790)	764 (320)	ND
Hanafusa	2017	mixed	mixed	142	ND	65.7	350	3709 (1730)	491 (320)	66.8 (32.4)
Ito	2015	mixed	CART	37	ND	59.7	100	3197 (1424)	302 (150)	93.1 (51.62)
Ito	2020	mixed	CART	43	ND	58.7	123	3207 (1427)	299 (152)	91.3 (53)
Iwaki	2018	mixed	KM-CART	19	ND	62.8	39	7000 (2600)	ND	ND
Kawata	2019	Gyn	CART	29	2	56.6	47	2937 (820)	272 (84)	85 (33.2)
Maeda	2014	Gas	CART	5	ND	63.6	51	4007 (1304)	561 (205)	75 (29.8)
Matsusaki	2020	mixed	KM-CART	2109	ND	60.7	2224 ^†^	6200 (2600)	610 (300)	67.3 (44.5)
Nagata	2020	Gas	CART	30	30	59.5	100	4000 (200)	ND	ND
Ohta	2017	Gas	CART	6	ND	73.8	12	3850	485	ND
Togami	2014	Gyn	NA	4	ND	ND	15	3190 (1086)	538 (249)	ND
Ueda	2012	Gyn	CART	22	14	ND	57	3290 (1200)	NA	ND
Wang	2015	Gyn	CART	9	6	67.7	58	7730 (3390)	920 (470)	161.2 (89.1)
Yamaguchi	2015	Gas	CART	30	30	58	127	3056 (1250)	334 (162)	85.5 (46.9)
Yamamoto	2021	Gyn	CART	31	11	66.4	49	3009 (1253)	392 (190)	ND

^†^: 4781 procedures conducted and data analyzed in 2224 procedures; Mixed: mixed kinds of malignancies; Gyn: gynecological malignancies; Gas: gastrointestinal malignancies; SD: standardized difference; CART: cell-free and concentrated ascites reinfusion therapy; KM-CART: Keisuke modified CART; ND: not described.

**Table 2 cancers-13-04873-t002:** Effect of CART on the alleviation of symptoms using a numerical rating scale system (0–10).

Symptoms	Before (95% CI)	After (95% CI)	Mean Difference (95% CI)	*p* Value
Abdominal distension	8.10 (7.78, 8.42)	2.12 (1.80, 2.44)	6.00 (5.49–6.51)	<0.01
Dyspnea	4.40 (3.03, 5.77)	1.67 (0.89, 2.45)	2.66 (2.05–3.28)	<0.01
Fatigue	6.17 (4.11, 8.23)	3.54 (2.27, 4.82)	2.64 (1.86–3.42)	<0.01
Lack of appetite	6.15 (4.56, 7.73)	3.62 (3.04, 4.20)	2.58 (1.53–3.63)	<0.01
Abdominal pain	3.90 (2.53, 5.27)	2.15 (1.26, 3.03)	1.74 (1.14–2.35)	<0.01
Nausea and vomiting	3.21 (1.05, 5.36)	1.79 (0.01, 4.14)	1.40 (0.86–1.95)	<0.01

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
