# Peer review of "Effectiveness of Cell-Free and Concentrated Ascites Reinfusion Therapy in the Treatment of Malignancy-Related Ascites: A Systematic Review and Meta-Analysis"

_cancers, 2021, doi:10.3390/cancers13194873_

Round 1

Reviewer 1 Report

This paper is a systematic review and meta-analysis of CART for malignant ascites. The findings of this paper will be important for understanding the efficacy and safety of CART. Although CART is still a local practice in Japan, the topic addressed is interesting and deserves a worldwide discussion. However, it still needs a considerable revision to be accepted for publication.

Major comments:

  1. Overall, it is important to emphasize that there is no standard treatment for malignant ascites because controlled trials to determine the efficacy and safety of each treatment including CART are lacking, and all studies included in this review were single arm studies which just compared some outcome measures before and just after CART procedures. In addition, there is no consensus about the optimal outcome measures of treatments for malignant ascites. Although symptom intensities or body weight is measured in a very short period, e.g., at the next day of CART, they just reflect the amount of removed ascites and are not likely to surrogate the effect of CART to control ascites reaccumulation or survival benefit. Thus, I believe that the efficacy and safety of CART cannot be confirmed in this study. I wonder the authors may over-interpret the results. Please consider the following suggestions:

In introduction, I suggest the authors to add “controlled trials to determine the efficacy and safety of CART are lacking” and “to explore the efficacy and safety of CART for malignant ascites, we conducted a systematic review and meta-analysis of observational studies.”

In limitation in Discussion, I suggest the authors to add that “there were no controlled trials.”

Please rephase the conclusion, abstract, and summary: e.g., “CART might be a safe and effetive…”, and “Further clinical trials are needed to confirm the efficacy and safety of CART for malignant ascites.”

  1. Risk of bias in Material and Methods. Please provide more details on Newcastle-Ottawa quality assessment. What and how is it measures. What do the scores mean? Additionally, please add a figure of the results of this assessment for the reviewers and readers to know the risk of bias in more detail.

  1. Outcomes in Material and Methods. Please clarify when the outcomes were measured, e.g., just after CART, the next day, or 1 week after CART. I believe we should know the time of the measurements to evaluate the efficacy of CART, or the appropriateness of the measurement.

  1. Outcomes in Material and Methods. Please clarify what is the primary outcome(s), and provide the reasons why you choose them. I wonder that the effects of reinfusion cannot be assessed with body weight or abdominal circumference just after the procedures or at the next day, because they will only reflect the amount of ascites removed. Similarly, other outcomes including PS, symptom intensities, and laboratory findings in a very short interval may not be valid to assess the efficacy of CART. Paracentesis intervals, on the other hand, might be affected by chemotherapy in this heterogeneous review. I recommend the authors to discuss about the optimal outcome measures in discussion and add in the limitation.

  1. Study search and study characteristics in Results and Table1. In each study, please clarify the number of patients undergoing chemotherapy, and whether the analyses performed per patients or per procedures. And please state in limitation if there might be some bias by analyzing per procedures.

  1. Please make sure whether the patients in Ishitani2021 is duplicated with those in Hanafusa2017, because Ishitani2021 seems to be a secondary analysis of the same study.

  1. Study search and study characteristics in Results and Table 1. Please explain why “Membrane” (CART or KM-CART) was important with appropriate citations. Additionally, please clarify how the membrane category confirmed for each study? I wonder some articles did not refer that. Did you contact the authors of each study?

  1. Study search and study characteristics in Results and Figure S6. Please clarify how the authors calculate the mean time to next paracentesis. Hanada2018 reported that the median time to next paracentesis was 27 days, but the authors reported in this manuscript that mean time to next paracentesis was 44.7 days in Hanada2018. Similarly, Ueda2012 reported that paracentesis interval was 44.7 days in CART with chemotherapy arm and 12.3 days in CART without chemotherapy arm, while the authors reported time to next paracentesis was 27 days. Furthermore, Hanafusa2017 and Maeda2014 did not report time to next paracentesis or paracentesis intervals but CART-CART intervals.

  1. Figures and Supplemental figures. Please indicate what the words mean in the legends, i.e., “Study or Subgroup”, “IV”, “Random”, “Favours [control]”, “Favours [CART]”, “Favours [experimental]”, and “Favours [After CART]”. Maybe the authors need modifications for these words and the horizontal axis of each figure to describe the content of each analysis.

Minor comments:

  1. Inclusion and exclusion criteria in Material and Methods. Please clarify why you included “conference abstracts”. Could there have been any concern about the quality? Did authors included them to lessen publication bias?

  1. Study search and study characteristics in Results. “All but two articles were reported in English”.

  1. Results. The word “most” is ambiguous, i.e., “KM-CART was used in most patients”, “PS was reported by most studies,” and “Most studies used 100 or 200mg of”. I suggest to modify them into exact numbers.

  1. Table2. Please provide each symptom intensity scores before and after the procedures for readers for better understanding of the effect on alleviation of symptoms.

I hope that my comment is very useful for the improvement of the article.

Author Response

Dear reviewer

We wish to express our strong appreciation to the reviewers for their insightful comments on our paper. We feel the comments have helped us significantly improve the paper. We strongly appreciate the reviewer read related references cited in this paper and pointed out mistakes in this paper. In particular, we wish to acknowledge their highly valuable comments on the Material and Methods section and conclusion section, which improved the quality of this paper. 

Major comment 1

Response: We strongly appreciate the reviewer's comment on this point. We agree with you and have incorporated this important suggestion throughout our paper. 

As the reviewer pointed out there are no controlled trials to determine the efficacy and safety of CART are lacking. We added “There was no controlled trial conducted between CART and puncture groups to determine the efficacy and safety of CART. To explore the efficacy and safety of CART for malignant ascites, we conducted a systematic review and meta-analysis of observational studies.” at the end of the introduction section. We also revised the contents in the conclusion as “CART might be a safe and effective palliative therapy for MRA. There were no serious adverse events during the procedures. Further clinical trials are needed to confirm the efficacy and safety of CART for malignant ascites.”

Major comment 2

Response: We wish to express our strong appreciation to the reviewers for their insightful comments on our paper. 

Due to the nature of single arm studies. We used a modified Newcastle-Ottawa quality assessment appraising the risk of bias. This content was added in the risk of bias section “Due to the nature of single arm studies, “selection of the non-exposed cohort” and “comparability” domains were not applicable in this study. The final score would then be 6 stars in max in modified Newcastle-Ottawa quality assessment”. We also added Sup Table 1 showed detailed information of scores in supplementary materials.

Major comment 3 & 4

Response: We strongly appreciate the reviewer's comment on this point. We agree with you and have incorporated this important suggestion throughout our paper. 

As the reviewer pointed, it is difficult to find a suitable primary outcome to identify the effect of CART due to limited data. The ratio of improved PS was considered as the long-term primary outcome. But there were only 4 articles that discussed improved PS as a category variable, other 6 articles analyzed PS as a continuous variable or without detailed information, which was not suitable for meta-analysis. 

There are three procedures during CART: paracentesis, concentrate, and reinfuse. We checked the short-term effect of CART by changes bodyweight, abdominal girth, and serum albumin. The outcomes were measured the next day of CART in each procedure. We added this content in the outcome section. “Laboratory findings were checked the day after each CART procedure and adverse events were appraised during the procedure of CART. The ratio of improved PS was considered as the primary outcome of the long-term effect of CART, but only four studies discussed PS as a category variable. A short-term effect of CART was analyzed by the changes bodyweight, abdominal girth, and serum albumin according to paracentesis and reinfuse procedures in CART.”

Major comment 5

Response: We appreciate the reviewer's comment on this point.

We added the number of patients who underwent chemotherapy after CART. The analysis of CART was conducted by each procedure, we added this information in the limitation section “Fifth, the effect of CART was analyzing per procedure, which posed a substantial risk of selection bias.”

Major comment 6

Response: We appreciate the reviewer's comment on this mistake.

We deleted the study data of Ishitani2021 and revised the contents of tables and figures in the manuscript accordingly.

Major comment 7

Response: We appreciate the reviewer's comment on this point. Our original expression here tended to be confusing. All studies used the same membrane described in result section. The correct label is type of CART.

Major comment 8

Response: We appreciate the reviewer's comment on this point. Our original expression here tended to be confusing.

Data from Hanada2018 and Ueda2012 should be changed. As the reviewer pointed out, we used data of CART with chemotherapy arm only from Ueda 2012. An average paracentesis interval was calculated from both the chemotherapy arm and without chemotherapy arm as 22.5 days. Data of Hanafusa2017 is available at the end of table 1. Paracentesis interval data Maeda2014 could be found at the beginning of the result section. They used words “session interval” instead of paracentesis interval, which made it hard to find out.

Major comment 9

Response: We appreciate the reviewer's comment on this point.

We modified “Study or Subgroup” to “Study” and the horizontal axis of each figure was also modified as necessary. “IV” and “Random” represented “inverse variance” and “random effect”, respectively. This is useful to understand the method of meta-analysis. We would like to keep it if possible.

Minor comment 1

Response: We appreciate the reviewer's comment on this point.

Actually, no conference abstract was enrolled in this meta-analysis. We deleted this content.

Minor comment 2

Response: We appreciate the reviewer's comment on this point.

We revised as “All but two articles were reported in English”

Minor comment 3 

Response: We appreciate the reviewer's comment on this point.

We modified “most” as exact numbers. The contents were as follows:

KM-CART was used in 2,128 patients”, “PS was reported by 10 studies,” and “nine studies used 100 or 200mg of”. 

Minor comment 4

Response: We appreciate the reviewer's comment on this point. symptom intensity scores before and after the procedures were added in table 2.

Thanks again for your treasure comments on this article, which made it more readable.

Best Regards.

Reviewer 2 Report

In this systematic review and meta-analysis the authors evaluated the effectiveness of cell-free and concentrated ascites reinfusion therapy in the treatment of malignancy-related ascites.

The article is methodologically accurate; nevertheless, I would like to suggest some improvements.

  • Abstract and Conclusion: "The effects of CART are potentially longer than those of total paracentesis alone." I would suggest to delete this statement in the abstract and conclusion as the comparison between CART and paracentesis was not the main aim of this study. Conversely, the authors should add in the abstract and in the conclusion (not only in the discussion) that all the included studies were from Japan and more studies are needed to generalize the results to the whole population.
  • Introduction and discussion sections could be enlarged.
  • Table 1: the column "Author" should be modified in "First author".

Author Response

Dear reviewer

We wish to express our strong appreciation to the reviewers for their insightful comments on our paper. We feel the comments have helped us significantly improve the paper. 

Major comment 1

Response: We appreciate the reviewer's comment on this point.

As the reviewer pointed we modified the contents of the abstract. “The effects of CART are potentially longer than those of total paracentesis alone.” was deleted and “Cell-free and concentrated ascites reinfusion therapy (CART) is one of the palliative treatments widely conducted in Japan only.” was added. 

Major comment 2

Response: We appreciate the reviewer's comment on this point.

We enlarged the contents in the introduction and discussion sections. 

This content was added in the introduction: “Hypoalbuminemia, which is prone to malignant ascites, has been reported as a risk factor for febrile neutropenia, a side effect of cancer chemotherapy. With the CART, with proteins included in ascites are collected and intravenously reinfused, avoiding the loss of beneficial proteins through paracentesis.

There was no controlled trial conducted between CART and puncture groups to determine the efficacy and safety of CART. To explore the efficacy and safety of CART for malignant ascites, we conducted a systematic review and meta-analysis of observational studies.” 

We also added content in the discussion section mainly in limitations.

Minor comment 1

Response: We appreciate the reviewer's comment on this point.

"Author" was modified in "First author".

Thank you for your useful suggestion. 

Best regards

Round 2

Reviewer 1 Report

This revised manuscript shows a massive effort to improve the quality of the article. However, I feel the authors have not adequately addressed all comments and many issues remain to be resolved before acceptance. Please examine the following points carefully.

Major comments:

  1. Risk of bias in Material and Methods.

I feel explanation of the Newcastle-Ottawa scale is still insufficient. Please explain what elements/items were originally included in the scale. Also, is there a rational reason why some of the items were excluded? Is your decision based on any formal criteria for using the scale? The selection of the non-exposed cohort and comparability are extremely important items to assess the risk of bias, and the fact that they are missing in all the studies included in this systematic review is essential information for assessing the quality of this review.

For the same reason, I suggest adding information on study design, comparison groups (or single arm), and primary and secondary endpoints to Table 1 to provide an overview of each study.

  1. Outcomes in Material and Methods.

The same comment should be noted. Please clarify when each outcome was measured. Was the performance status evaluated just after CART, the next day, or 1 week after CART? When the symptoms were measured? I believe the timepoint of the measurements is important to assess the reliability of the outcome investigated: outcomes just after CART may not surrogate the true endpoint, e.g., overall survival, QOL, ascites volume reduction, or symptom control.

  1. Outcomes in Material and Methods.

I agree with the authors comment that long-term PS was a good candidate for the primary outcome. Please clarify that in the manuscript. Then, the result of this systematic review might be “no relevant paper found,” and the conclusion might be “no evidence about whether CART improved PS or not,” or “Although various outcomes were compared before and just after the procedure, they seemed lack reliability for the efficacy endpoint of CART. No study investigated long term effect of CART. Thus, a controlled trial with a validated outcome is warranted.

This is a critical point. Please consider carefully. I recommend the authors to refer the Clinical Guideline for Gastric Cancer 6th ed (Japanese Gastric Cancer Association, Kanehara, 2021) in which evidence of CART for malignant ascites are well summarized.

  1. Study search and study characteristics in Results and Table 1.

Please explain why the type of CART (original or KM-CART) was important with citations for readers in the world. Additionally, please clarify how the types confirmed for each study? I wonder some studies might use KM-CART although they did not refer that in the article. Did you contact the authors of each study?

  1. Study search and study characteristics in Results and Figure S6.

Please clarify whether the authors reported “median” or “mean” time to next paracentesis. Please fix the word in the title of the figure S6 or the manuscript. Then, please clarify how the authors calculate the “mean” (or median) time to next paracentesis. Hanada2018 reported the “median” time to next paracentesis, while Ueda2012 reported the “mean” paracentesis interval (44.7 days in CART with chemotherapy arm and 12.3 days in CART without chemotherapy arm).

  1. Hanafusa2017 and Maeda2014 did not report time to next paracentesis or paracentesis intervals but CART-CART intervals. CART-CART intervals are not equal to time to next paracentesis when the next procedure is simple paracentesis, or the patients were censored. Including CART-CART intervals in the time to next paracentesis may be a considerable risk of bias.

Minor comments:

  1. Figures and Supplemental figures.

In Figure S2, for example, I suggest the authors to change the word “mean difference” to “mean volume of ascites collected (L).”

I hope that my comment is very useful for the improvement of the article.

Author Response

Dear reviewer

We wish to express our strong appreciation to the reviewers for their insightful comments on our paper. We feel the comments have helped us significantly improve the paper.

Major comment 1

Response: Response: We appreciate the reviewer's comment on this point. As the reviewer pointed out: this meta-analysis was conducted on single-arm studies. Due to the nature of single-arm study, “non-exposed cohort” and “comparability” are usually deleted in Newcastle-Ottawa scale. A recently published paper “Contemporary Neoadjuvant Therapies for High-Risk Melanoma: A Systematic Review” (https://doi.org/ 10.3390/cancers13081905) used the same modified Newcastle-Ottawa scale appraising single-arm studies.

As for outcomes of enrolled studies. Only one study clarified the primary and secondary outcomes of the study. Primary outcomes in other studies were not clear. Due to the limited space in table 1, we created a Sup table 2 summarizing information of outcomes.

Major comment 2

Response: We strongly appreciate the reviewer's comment on this point. We agree with you and have incorporated this important suggestion throughout our paper. Symptoms were evaluated the next day after CART. PS was appraised the next day in studies of Ohata 2017 and Ueda 2012. The study of Hagata 2020 described appraising PS and laboratory findings immediately after CART. They explained check L/D within two days after CART, but detailed information of timing in appraising PS was unclear. Within one day or two days seemed no obvious difference from a long-term view. We analyzed data together. This information was added in the methods section:“Symptoms were compared before and the next day after CART. PS was rechecked soon after CART.”

Major comment 3

Response: We strongly appreciate the reviewer's comment on this point. We agree with you and have incorporated this important suggestion throughout our paper. We added this information to the discussion. “The limited study investigated the long-term effect of CART. Although various outcomes were compared before and just after the procedure, they seemed to lack reliability for the long-term efficacy of CART.” In the limitation section.

Major comment 4 

Response: We appreciate the reviewer's comment on this point. Our original expression here tended to be confusing. If information on KM-CART can’t be confirmed from the article, then CART was generally considered. We didn’t contact the authors of each study to confirm the type of CART, because this meta-analysis was not designed to compare different effects between CART and KM-CART.

Major comment 5

Response: We strongly appreciate the reviewer's comment on this point. The data of mean time to next paracentesis with SD from Hanada 2018 and Ueda 2012 were calculated from Figure 1a and Figure 4, respectively. We read the values of days in each procedure in the figures and then calculated mean and SD. We added a notation of “calculated from figures” of Sup Figure 6 to make it clear.

Major comment 6

Response: We appreciate the reviewer's comment on this point. Our original expression here tended to be confusing. As the reviewer pointed out paracentesis to paracentesis intervals were different from CART-CART intervals. The effect of discontinued CART was unclear and not discussed by the authors. Probably they were excluded by studies. We added a notation of Sup Figure 6 to Hanafusa 2017 and Maeda 2014 to clarify the differences.

Minor comment 1

 Response: We appreciate the reviewer's comment on this point. “mean differences”, “IV”, “Random (Effect)” are fixed output expressions in software Review Manager. It showed the method of calculation in meta-analysis. We modified figures make it easy to understand.

Thanks again for your comment on the improvement of the article. 

Reviewer 2 Report

The authors revised the manuscript taking into account the comments of the reviewers

Author Response

Dear reviewer

We wish to express our strong appreciation to the reviewers for their insightful comments on our paper. We feel the comments have helped us significantly improve the paper.

Thanks again